# Postoperative Electroacupuncture Boosts Cognitive Function Recovery after Laparotomy in Mice

**DOI:** 10.3390/biom14101274

**Published:** 2024-10-10

**Authors:** Yuen-Shan Ho, Wai-Yin Cheng, Michael Siu-Lun Lai, Chi-Fai Lau, Gordon Tin-Chun Wong, Wing-Fai Yeung, Raymond Chuen-Chung Chang

**Affiliations:** 1School of Nursing, Faculty of Health and Social Sciences, The Hong Kong Polytechnic University, Hong Kong SAR, China; siu-lun-michael.lai@polyu.edu.hk (M.S.-L.L.); jefferyl@hku.hk (C.-F.L.); jerry-wf.yeung@polyu.edu.hk (W.-F.Y.); 2Laboratory of Neurodegenerative Diseases, School of Biomedical Sciences, LKS Faculty of Medicine, The University of Hong Kong, Hong Kong SAR, China; wai-yin-nano.cheng@polyu.edu.hk; 3Research Institute for Future Food, The Hong Kong Polytechnic University, Hong Kong SAR, China; 4Department of Food Science and Nutrition, Faculty of Science, The Hong Kong Polytechnic University, Hong Kong SAR, China; 5Department of Anesthesiology, The University of Hong Kong, Hong Kong SAR, China; gordon@hku.hk

**Keywords:** cognition, memory, surgery, acupuncture, Chinese medicine, tau, inflammation

## Abstract

Postoperative cognitive dysfunction (POCD) is a common complication that affects memory, executive function, and processing speed postoperatively. The pathogenesis of POCD is linked to excessive neuroinflammation and pre-existing Alzheimer’s disease (AD) pathology. Previous studies have shown that acupuncture improves cognition in the early phase of POCD. However, POCD can last for longer periods (up to weeks and years). The long-term effects of acupuncture are unknown. In this study, we hypothesized that electroacupuncture (EA) could reduce inflammation and cognitive dysfunction induced by laparotomy over a longer period. We characterized the effects of postoperative EA on cognitive changes and investigated the underlying molecular mechanisms in mice. Laparotomy was performed in 3-month-old mice followed by daily EA treatment for 2 weeks. Our data indicated that laparotomy induced prolonged impairment in memory and executive functions, which were mitigated by postoperative EA. EA also reduced tau phosphorylation and suppressed the activation of tau-related kinases and glia, with effects comparable to ibuprofen. These findings demonstrate the beneficial effects of EA in a mouse model of POCD, suggesting that EA’s ability to suppress neuroinflammation may contribute to its protective effects. In conclusion, EA may be a viable non-pharmacological intervention for managing POCD in different phases of the medical condition.

## 1. Introduction

Perioperative care is important for promoting patient recovery and improving quality of life. After general anesthesia and surgery, some patients experience cognitive dysfunction, which differs from postoperative delirium. This kind of cognitive change is called postoperative cognitive dysfunction (POCD). POCD can affect multiple cognitive domains, including attention, concentration, executive function, memory, visuospatial ability, and psychomotor speed, over an extended period of time, ranging from weeks to years, as reported in studies [1,2,3]. The impact of POCD can be profound, affecting both patients and their families. In particular, patients with POCD at hospital discharge (i.e., around 7 days) [4] or 3 months have been associated with increased mortality [5].

Although no definitive pathophysiological mechanism for POCD has been identified, it is widely believed that this condition is related to multiple factors. Potential contributing factors to POCD include inflammatory responses triggered by surgical procedures, excessive neuronal cell death, heightened oxidative stress, free radical damage, and alterations in synaptic function [6]. Inflammation is thought to play a significant role in POCD, with both systemic and neuroinflammation triggered by peripheral surgical trauma or anesthesia, and has been proposed as a cause of the observed cognitive deficits [7,8,9,10]. Elevated levels of pro-inflammatory cytokines such as IL-6, IL-1β, and TNF-α have been reported after surgery and may be related to POCD [10]. Dysfunction of the blood–brain barrier (BBB) may also facilitate the transduction of systemic inflammatory cytokine responses to the central nervous system. During inflammatory conditions, the permeability of the BBB increases, allowing more inflammatory cytokines and cells, such as macrophages, to enter the CNS [11]. Inflammation in the brain can interfere with long-term potentiation (LTP), memory consolidation, and neurogenesis, thereby affecting memory and learning [12]. 

Microglia play a crucial role in inflammatory processes. After surgery, changes in microglia morphology and microglia activation occur [13]. Activated microglia are more likely to produce a strong response to peripheral stimuli caused by stress or surgical trauma. Microglia can trigger cognitive dysfunction through various signaling pathways [14,15]. They can impair synaptic function by engulfing synapses in the hippocampus, ultimately resulting in synaptic loss [16]. Notably, POCD exacerbates the long-term risk of Alzheimer’s disease (AD) [3]. Our group has previously reported on Alzheimer’s disease-related pathology and neuroinflammation in the brains of mice following laparotomy. These effects were partially suppressed by the anti-inflammatory drug ibuprofen [10], suggesting that inflammation plays a significant role in the development of POCD.

In the medical context, a non-pharmacological approach for POCD is frequently preferred because various medications may have adverse effects on patients, particularly those with comorbidities, such as the elderly. For instance, non-steroidal anti-inflammatory drugs administered during the postoperative period for pain relief can lead to gastric bleeding and renal damage, rendering them unsuitable for long-term use. Moreover, there are currently no approved medications for the treatment of POCD, although some clinical reports suggest that the use of lidocaine (a local anesthetic) or parecoxib (a nonsteroidal anti-inflammatory drug) may reduce the incidence of POCD [17,18]. Therefore, it is essential to investigate and develop effective management strategies to help patients restore their cognitive function at different phases after surgery.

Acupuncture is a significant technique in traditional Chinese medicine. It involves inserting thin needles into specific acupoints on the body and manipulating them manually or applying electrical stimulation (known as electroacupuncture or EA) to stimulate various physiological processes [19]. The incorporation of acupuncture in clinical settings is becoming increasingly popular due to its potential to enhance cognitive function without the risk of the adverse effects associated with pharmacological interventions. Clinical and laboratory investigations have suggested that acupuncture administered prior to surgery may help reduce the incidence of POCD and suppress systemic inflammation [8,20]. Additionally, the incidence of agitation and delirium were also reduced in those who received acupuncture treatment [20,21,22]. However, as acupuncture is more likely to be utilized after surgery in most situations, additional scientific investigation is required in order to demonstrate its effectiveness. Furthermore, most pre-clinical studies only investigated the pathogenesis and interventions for POCD in its early phase (usually within a week); there is limited understanding of how short-term or long-term POCD can be managed and the pathological changes involved. There is substantial evidence supporting the use of acupuncture for managing cognitive impairment in various cases. For example, acupuncture used alone or in combination with other therapies can enhance cognitive function and daily living skills in patients diagnosed with vascular dementia [23]. It has also been suggested to improve cognition and reduce fatigue in patients experiencing long COVID [24]. Notably, acupuncture and related therapies are effective in alleviating postoperative pain and promoting tissue recovery [25,26,27]. Given that pain and cognitive impairment frequently coexist in patients after surgery, interventions that target both are particularly attractive to physicians.

In this study, we hypothesized that electroacupuncture (EA) could be used to mitigate inflammation during laparotomy-induced prolonged cognitive dysfunction and Alzheimer’s disease (AD)-related pathological changes. To test this hypothesis, we established the following objectives: (1) to characterize the prolonged effects of postoperative EA in mice following laparotomy under general anesthesia, with a focus on cognitive changes, and (2) to explore the molecular mechanisms underlying the observed EA-mediated changes in inflammation and AD-related pathology after laparotomy.

## 2. Materials and Methods

### 2.1. Animal Husbandry

Twelve-week-old male C57BL/6N mice, weighing 25 ± 3 g, representing early adulthood, were purchased from the Center for Comparative Medicine Research, University of Hong Kong. Only male mice were used in this study because gender differences in spatial learning, memory, and hippocampal plasticity have been previously reported [28,29,30]. In this pre-clinical study, we aimed to minimize the influence of gender factors and focus on the therapeutic effects of acupuncture. Therefore, we selected male mice as the study subjects. All animals were housed in the Center for Comparative Medicine Research, which is fully accredited by the Association for Assessment and Accreditation of Laboratory Animal Care International (AAALAC). Mice were kept in a temperature-controlled room at 20–22 °C with a humidity of 50 ± 10% and a 12/12 h light/dark cycle. The mice had access to food and water ad libitum. Mouse handling and all other procedures were carried out in compliance with the National Institutes of Health’s Guide for the Care and Use of Laboratory Animals and the Animals (Control of Experiments) Ordinance in Hong Kong, China. The use of animals was approved by the Department of Health in Hong Kong, the Committee on the Use of Live Animals in Teaching and Research at The University of Hong Kong (#4516-17, 18 January 2018), and the Animal Subjects Ethics Sub-Committee at The Hong Kong Polytechnic University (#17-18/26-SN-R-HMRF, 6 November 2017). Every effort was made to reduce the number of animals used and to minimize suffering. Mice were coded so that the person who performed the data analysis was blinded to their identity.

### 2.2. Grouping of Animals

In the initial phase of our experiment, we examined the consequences of postoperative electroacupuncture (EA) in mice that had undergone laparotomy under general anesthesia. The mice were randomly divided into three groups: (1) sevoflurane alone (control), (2) laparotomy alone, and (3) laparotomy plus EA. In the subsequent section of our study, we compared the effects of EA and ibuprofen (IBU). Mice were randomly divided into two groups: (1) laparotomy plus EA and (2) laparotomy plus ibuprofen (positive control). Based on our electroacupuncture protocol described in Section 2.4, mice were entered into either the 7-day or 14-day protocol and were sacrificed on day 7 or day 14, respectively.

### 2.3. Surgery

In this study, we used a laparotomy mouse model of POCD. Laparotomy was performed according to our previously published paper [10]. The animals were anesthetized for 20 min. Initially, the mice were placed in a chamber with 5% sevoflurane (Sevorane™, Abbott, Basel, Switzerland) and subsequently transferred to a nose mask with anesthesia maintained at 3% sevoflurane and a gas flow of 800 mL/min using an inhalation anesthesia machine (Harvard Apparatus, Holliston, MA, USA). The control group received no further intervention, whereas the laparotomy group underwent surgical intervention. Prior to laparotomy, hair was removed from the surgical site, and the skin was disinfected. A 2 cm longitudinal midline incision was made at a distance of 0.5 cm below the lower right rib. A segment of the small intestine approximately 10 cm in length was temporarily exteriorized and subjected to gentle manipulation for 2 min before being placed back into the abdomen. The abdominal muscles and skin were sutured using 5–0 vicryl and nylon sutures (Ethicon, Somerville, MA, USA). The surgical procedure was completed within 20 min, and the respiration frequency and rhythm, as well as the color of the paw, were closely monitored. Upon recovery from anesthesia, the mice were allowed to move and reorient for 2 min before returning to their home cages. Analgesia was administered for 3 days following laparotomy.

### 2.4. Electroacupuncture (EA) Protocol

Mice in the acupuncture group underwent electroacupuncture (EA) treatment after laparotomy. EA treatment was initiated on day 2 to provide sufficient rest for mice following the surgical procedure. A custom-made nylon net was used for physical restriction to ensure that the mice remained in place during treatment. Acupuncture needles (0.18 × 13 mm, Hwato) were inserted horizontally at the Baihui (DU20) and Zusanli (ST36) points, which were selected according to a published review paper [8]. The Baihui point is located at the intersection of the sagittal midline and the line linking the two ears, at a depth of 2–3 mm. The Zusanli point is situated approximately 2 mm below the fibular head on the posterolateral knee of the hind limbs, bilaterally, at a depth of 3–4 mm. An EA apparatus (Hwato; Suzhou Medical Instruments Co., Ltd., Suzhou, China) was used to generate waves at 1 and 20 Hz. The 7 days and 14 days protocols were used, with EA treatment lasting for 20 min daily for 5 or 12 days, respectively. These time points were chosen because they correspond to the early and middle postoperative periods and have been reported to be effective in improving cognitive function [8,31]. The control, laparotomy-only, and ibuprofen groups were subjected to similar physical restrictions using an identical apparatus, but without EA. To ensure the integrity of the study, behavioral tests were conducted prior to EA treatment if EA and behavioral tests were performed on the same day.

### 2.5. Ibuprofen (IBU) Treatment

The oral administration of ibuprofen (Sigma-Aldrich, St. Louis, MO, USA) (60 mg/kg/day) was given to the mice in the ‘laparotomy + ibuprofen’ group for 5 or 12 days following the surgery.

### 2.6. Body Weight Measurements

Body weight was monitored weekly until the end of the study using a standard scale, and the percentage change from the initial measurement was reported.

### 2.7. Behavioral Tests

#### 2.7.1. Novel Object Recognition (NOR) Test

The ability of rodents to recognize a new object was assessed using a Novel Object Recognition (NOR) task in a controlled setting [32]. In this task, the mice were placed in an open-field arena for habituation for 24 h without any objects. Subsequently, they were placed in the same arena with two identical sample objects (A + A) and allowed to explore the environment for a retention interval of 24 h. Finally, the animals were returned to the arena with two objects, one identical to the sample and the other novel (A + B), and the time spent exploring each object was recorded to calculate the discrimination index, which was used to evaluate recognition memory as the ratio of the time spent exploring one object to the time spent exploring two objects.

#### 2.7.2. Open Field Test

General locomotor activity and anxiety levels of mice were assessed using the open field test, following a previously described protocol [33,34]. Each mouse was placed in the center of an empty 40 × 40 cm square arena and allowed to wander freely for 10 min, during which time its behavior was recorded using a video camera mounted overhead. The data were analyzed using video tracking software (SMART 3.0, Panlab SL, Barcelona, Spain), which divided the arena into 25 zones (16 peripheral and 9 central). The total distance traveled and the duration of time spent in the center and outer zones were measured.

#### 2.7.3. Y-Maze Test

The associated memory was assessed using the Y-maze test. Individual mice were placed in an apparatus consisting of three connected arms. On day 1 (learning trial), an individual mouse was placed in the center of the Y-maze and allowed to move freely through the maze for 5 min. During this time, electrical shocks (2 Hz, 125 ms, 10 V) were delivered through the grid floor in the two compartments. These compartments were dark, whereas the shock-free compartment was brightly lit. The correct choice refers to entering the shock-free compartment and remaining there for 30 s. Each mouse was trained ten times. On day 2 (testing trial), each mouse was tested 10 times, following the same procedure as on day 1. The number of incorrect choices on day 2 was recorded.

#### 2.7.4. Puzzle Box Test

Problem-solving ability, short-term memory, and long-term memory were assessed by the puzzle box test using a 4-day protocol modified by our laboratory [32] based on a previously published method by O’Connor et al. [35]. The testing arena was composed of two compartments separated by a removable barrier: a bright-lit start zone (600 mm × 280 mm) and a smaller covered goal zone (600 mm × 280 mm). Briefly, each mouse was initially introduced into the start zone and trained to move to the goal zone through a narrow underpass. Mice were subjected to 10 trials (T1–T10) over four consecutive days, with three trials on the first three days and one trial on the last day. During this period, the entrance to the goal zone was blocked by different objects with increasing difficulty. Each obstruction included three trials; the first two took place on the same day, with the third trial conducted the following day. The time required for each mouse to enter the goal zone was recorded. Trials 2, 5, and 8 were used to detect problem-solving ability. Trials 3, 6, and 9 were used to assess short-term memory. Trials 4, 7, and 10 were used to detect the long-term memory. The timeline of the experimental procedure is illustrated in Figure 1.

### 2.8. Biochemical Tests

The mice were humanely euthanized following the conclusion of their behavioral tests. Subsequently, their plasma and brains were collected, and the brains were sectioned into the frontal cortex and hippocampus.

#### 2.8.1. Total RNA Extraction, Reverse Transcription, and Real-Time PCR

Total RNA was extracted from the hippocampus and frontal cortex using TRI Reagent^®^. Mice were euthanized via carbon dioxide asphyxiation, and tissues from the hippocampus, frontal cortex, and plasma were rapidly collected and snap-frozen in liquid nitrogen. Brain tissues were mechanically homogenized under RNase-free conditions using RNAiso plus (Takara, Shiga, Japan) and resuspended in DEPC-treated water. RNA quality was assessed with Nanodrop One (Thermo Fisher Scientific, Waltham, MA, USA), ensuring A260/A280 ratios between 1.8 and 2.0. Subsequently, 0.5 μg of RNA was used for reverse transcription with the PrimeScript^TM^ Master Mix Kit (Perfect Real Time) (Takara, Japan), and cDNA samples were diluted tenfold in DEPC-treated water.

Five microliters of diluted cDNA were amplified in triplicate using the TB Premix Ex TaqTM II kit (Takara, Shiga, Japan) and the CFX96 Touch Two-Color Real-Time PCR Detection System (Bio-Rad, Hercules, CA, USA). Amplification conditions were as follows: 95 °C for 30 s, followed by 41 cycles of 95 °C for 5 s, and 60 °C for 30 s. The relative expression levels of each cytokine were normalized to the housekeeping gene GAPDH using the 2^−ΔΔCt^ method. Primer sequences are listed in Appendix A.

#### 2.8.2. Brain Tissue Protein Extraction and Western Blot Analysis

Mouse brain tissues were homogenized in a lysis buffer containing RIPA buffer (Thermo Fisher Scientific, Waltham, MA, USA) with protease inhibitor cocktail (Sigma Aldrich, St. Louis, MO, USA), and a phosphatase inhibitor cocktail (Millipore, Bedford, MA, USA). The homogenates were then centrifuged at 20,000× *g* and 4 °C for 20 min, and the supernatants were collected. Synaptosomal fractions were freshly prepared using Syn-PER™ Synaptic Protein Extraction Reagent (Thermo Fisher Scientific, Waltham, MA, USA) with protease and phosphatase inhibitors, according to the manufacturer’s instructions.

The extracted proteins were subjected to electrophoresis on 10% polyacrylamide gel. Membranes were blocked with 5% non-fat dry milk and incubated overnight at 4 °C with primary antibodies (Appendix A). Horseradish peroxidase-conjugated secondary antibodies (DAKO, Glostrup, Denmark) were then used. The signal intensity of the immunoreactive bands was visualized by chemiluminescence (ECL or ECL-plus, Amersham GE Healthcare, Little Chalfont, UK). All immunoblots were normalized for gel loading by using antibodies against β-actin, GAPDH, or α-tubulin. Band intensities were measured using the ImageJ software (version 1.53t, National Institutes of Health, Bethesda, MD, USA).

#### 2.8.3. Immunohistochemical (IHC) Staining

Whole brains were surgically removed after being perfused with saline and then fixed with 4% paraformaldehyde. The Accu-OPTIClear tissue clearing method was used to remove lipids, achieving optical transparency [36]. The tissue was dehydrated, embedded in paraffin, and sectioned to study molecular changes in glial fibrillary acidic protein (GFAP) and ionized calcium-binding adaptor molecule 1 (Iba-1) for neuroinflammation. Immunohistochemical staining using the avidin–biotin complex (ABC) technique confirmed the findings. Slides were incubated with 3% H_2_O_2_ to inhibit endogenous peroxidase activity, then with 10% normal goat serum, followed by antibodies against GFAP and Iba-1. After washing, the slides were treated with a biotinylated secondary antibody, followed by ABC solution (Vectastain ABC Elite kit), and incubated with DAB substrate. GFAP and Iba-1 staining images were captured using a Leica microscope (CTR5000, Leica Camera AG, Wetzlar, Germany).

#### 2.8.4. Milliplex Cytokine Assays

The concentrations of interleukin-1β (IL-1β), tumor necrosis factor (TNF-α), Monocyte Chemoattractant Protein-1 (MCP-1), and IL-6 and IL-10 proteins in plasma were determined using a customized Milliplex Mouse Cytokine Immunoassay Kit and Analyzer 3.1 Luminex 22 machine (Millipore, Billerica, MA, USA), following the manufacturer’s instructions.

### 2.9. Statistical Analysis

The number of mice in each treatment group was calculated using the following equation: N = 1 + 2C(s/d)^2^, from the Institute for Laboratory Animal Research (U.S.) Committee on Guidelines for the Use of Animals in Neuroscience and Behavioral Research [Power (1-β) of the treatments = 80%; significant level α = 5%, C = 7.85] [37]. All data were initially assessed for normal distribution using the Shapiro–Wilk normality test. Statistical analysis was conducted using either an unpaired *t*-test, Mann–Whitney test, one-way analysis of variance (ANOVA) with Bonferroni’s post hoc test, the Kruskal–Wallis test with Dunn’s post hoc test, or two-way ANOVA with Bonferroni’s post hoc test, wherever applicable. All results are expressed as mean ± SEM. *p* < 0.05 was considered to be statistically significant for all analyses. All statistical analyses were performed using Prism (version 8.0, GraphPad Software, San Diego, CA, USA).

## 3. Results

### 3.1. Laparotomy Led to Notable Weight Loss in Mice without Affecting General Locomotor Activity or Anxiety- and Depression-like Behaviors

To assess the consequences of laparotomy and EA on the overall well-being of the mice, we monitored their body weight on an alternative schedule following surgery. Our observations revealed that laparotomy resulted in significant weight loss in mice that could not be reversed by EA. Locomotor activity was assessed by measuring the total distance traveled in the arena during the open field test. Anxiety- or depression-like behavior was gauged by the amount of time spent in the outer zone during the open field test. It is expected that mice exhibiting increased anxiety levels would spend more time in the outer zone. There was no significant difference between the three groups in terms of the total distance traveled and time spent in the outer zone, indicating no differences in anxiety- or depression-like behavior (Appendix A).

### 3.2. EA Attenuated Cognitive Impairment Induced by Laparotomy

In order to assess the short- and medium-term impacts of EA on cognitive function, cognitive performance was evaluated at different time points in the 7-day (upper panel) and 14-day protocols (lower panel) (Figure 2; for a schedule of the cognitive tests, please refer to Figure 1). In the Y-maze test, mice in the LAP group exhibited a significantly higher number of errors than the control mice at both the early and medium time points. However, those that received EA demonstrated improved performance in the Y-maze test, indicating an enhancement in associated memory (Figure 2a,d). In the novel object recognition test, we observed that mice that underwent laparotomy struggled to differentiate between the novel object and the old object, suggesting a deficit in recognition memory. Notably, mice that received EA for 12 days showed significant improvement in the novel object recognition test (Figure 2f).

### 3.3. EA Attenuated Laparotomy-Induced Tau Phosphorylation and Associated Kinases

To evaluate whether acupuncture could mitigate the pathological changes associated with Alzheimer’s disease, we conducted Western blot analysis to assess any alterations in tau phosphorylation. As depicted in Figure 3a,b, laparotomy induced the phosphorylation of tau (AT180, AT8, Ser404, and Ser396) in the hippocampus and frontal cortex of mice 14 days after surgery. This observation was consistent with our previous findings, demonstrating that the change in tau phosphorylation is not a transient one but appears to persist in the medium phase of postoperative cognitive dysfunction [10]. These changes, to some extent, were attenuated by acupuncture. However, the effects of acupuncture on reducing tau phosphorylation varied slightly between the hippocampus and the frontal cortex.

The degree of phosphorylation of the tau protein is contingent upon the equilibrium between the activity of kinases and phosphatases (Figure 4). Our findings indicate that these modifications emerge in the early period following surgery (7 days postoperatively). Our results showed that electroacupuncture (EA) diminished the activation of GSK3β and decreased the phosphorylation of JNK following laparotomy in both the hippocampus and frontal cortex. However, there were no observable changes in the JAK/STAT signaling proteins (Figure 4).

### 3.4. EA Did Not Produce a Noticeable Alteration in the Proteins Associated with Synaptic Transmission

Synaptic proteins, such as synapsin-1, synaptophysin, and NMDA receptor 2B (NMDAR2B), play a pivotal role in neurotransmission. We hypothesized that EA affects brain expression, thereby leading to cognitive improvement. To investigate this, we conducted Western blot analysis to assess changes in the expression of these synaptic proteins in the hippocampus and frontal cortex 14 days after laparotomy. Our results indicated that there were minimal changes in the expression of synaptic proteins in both regions. Specifically, we found that only the levels of NMDAR2B were significantly reduced in the frontal cortex (Figure 5b), whereas the expression of synapsin-1 and synaptophysin in the frontal cortex remained unchanged. The levels of NMDAR2B in the frontal cortex were not increased by EA (Figure 5b). Moreover, there were no changes in the expression of these proteins in the hippocampus (Figure 5a).

### 3.5. Inflammatory Response in the Central Nervous System after Laparotomy Was Suppressed by EA

Surgical interventions have been found to trigger immediate inflammation in peripheral regions and activate an immune response in the central nervous system for a considerable period following the procedure. To investigate the changes in central nervous system inflammation, the mRNA expression of inflammatory cytokines was analyzed in the hippocampus and frontal cortex of mice 7 and 14 days after laparotomy. As illustrated in Figure 6 there were no significant fluctuations in the levels of cytokines, including IL-1β, TNF-α, MCP-1, IL-6, and IL-10, across all groups at either time point. However, 14 days after laparotomy, a slight yet statistically significant increase in TNF-α (from 1.001 ± 0.063 to 1.355 ± 0.190, *p* = 0.0024) and IL-10 (from 1.115 ± 0.059 to 1.802 ± 0.171, *p* = 0.0005) levels was observed in the hippocampi of mice in the laparotomy group. Additionally, EA reduced elevated levels of IL-10 (1.345 ± 0.092, *p* = 0.0245) (Figure 6c).

To better comprehend the consequences of electroacupuncture (EA) on neuroinflammation, we analyzed variations in Iba-1 and GFAP levels within the hippocampus of mice. To this end, we used two distinct staining methods. Initially, we employed the Accu-OPTIClear tissue-clearing technique for tissue preparation, which allowed for a more precise visualization of the structure. Our findings revealed that the immunoreactivity of Iba-1, a marker for microglia, and GFAP, a marker for astrocytes, was elevated in the hippocampus of mice on day 14 after laparotomy. These results indicated the presence of neuroinflammation in the affected brain region. However, it is noteworthy that this increased immunoreactivity was mitigated in the mice treated with EA (Figure 7a). Our results were further validated by conducting immunohistochemical staining on brain slides using the conventional ABC immunohistochemical staining method (Figure 7b).

### 3.6. The Protective Effects of EA Were Compared to Those of Ibuprofen

Ibuprofen is a non-steroidal anti-inflammatory drug that is commonly used to manage postoperative pain. Our previous research demonstrated that ibuprofen suppresses inflammation and improves cognitive function after surgery [10]. In this study, we compared the effects of electroacupuncture (EA) and ibuprofen on cognitive function in mice. To assess the general physical condition of the mice in both groups, we measured their body weights and found no difference between the two groups (Appendix A). We then evaluated the performance of the mice in the Y-maze, NOR, and puzzle box tests on days 6 and 14 post-surgery (Figure 8 and Figure 9). On day 6 post-surgery, mice in the EA group performed worse than those in the ibuprofen group in the Y-maze test (Figure 8a), but their performance in the NOR test was similar (Figure 8c). On day 14 post-surgery, there was no significant difference between the two groups in the Y-maze test (Figure 8d); however, the ibuprofen-treated mice performed significantly better than the EA mice in the NOR test (Figure 8f). These results indicate that EA had a positive effect on cognitive function after surgery. To further investigate the potential benefits of EA, we used the puzzle-box test to study executive function in mice. We found that the problem-solving ability, short-term memory, and long-term memory of mice that underwent laparotomy were significantly impaired during the middle postoperative period, and these cognitive changes were attenuated by both EA and ibuprofen. No significant difference was observed between the EA and ibuprofen groups in the puzzle box test (Figure 9).

## 4. Discussion

Postoperative cognitive dysfunction is an important complication that requires further investigation. Acupuncture administered during the perioperative period may be a valuable strategy to mitigate the impact of surgery on cognitive function. In this study, a clinically relevant mouse model of laparotomy was employed to induce prolonged cognitive impairment and assess the effects of post-surgery EA on cognitive function. Changes in various cognitive domains were analyzed to provide a comprehensive understanding of the potential benefits of EA in POCD. By extending the examination period to two weeks, we evaluated the impact of EA on the development of AD-like pathology.

The laparotomy model is a suitable and relevant paradigm for studying systemic inflammation-triggered neuroimmune responses and cognitive dysfunctions. The surgical procedures were performed under sterile conditions and did not involve any ischemia/reperfusion [38,39], which induced drastic elevation of free radicals in peripheral circulation. Therefore, it mimics the conditions of many surgeries [10]. Most importantly, the laparotomy model induces prolonged cognitive impairment for up to 2 weeks, which allows the study of interventions and their mechanisms at later stages of POCD.

This study aimed to investigate the effects of postoperative EA on POCD. Our analysis revealed changes in behavior and pathology during the early (7 days) and middle (14 days) postoperative periods. Our results suggest that postoperative electroacupuncture (EA) can mitigate cognitive impairment after laparotomy. However, this effect was not apparent in the early postoperative period, during which EA only improved performance in the Y-maze test but not in the NOR test (Figure 2a,c). This result contrasts with other studies that suggest acupuncture improves cognitive function in the early post-surgery period (around 7 days after surgery) [38,40,41,42,43,44,45]. It is worth noting that in most of these studies, EA treatment began before surgery and continued for one week after the operation. In contrast, in our study, EA was initiated on the second postoperative day. We contend that this difference in treatment timing may account for the disparities in the results. 

POCD is present in approximately one-third of patients and its prevalence increases with age [4,46]. Acupuncture is not a widely available in-hospital service option in most countries, which makes it less practical to provide all patients with preoperative acupuncture as prophylactics to reduce perioperative complications including POCD. This diminishes the reference value of the widely studied presurgical acupuncture protocol for POCD. We believe that it is important to have more evidence to help clinicians to determine if EA given after surgery can still be effective. Although our data support the use of post-surgical acupuncture for POCD, it appears to require a greater number of EA treatments to achieve therapeutic effects. Five EA treatments (as used in our 7-day protocol) are unable to provide full protection to attenuate the cognitive impairment, whereas 12 treatments (as used in our 14-day protocol) effectively attenuated POCD. Interestingly, some researchers reported that even a session of EA or EA during surgery is sufficient to reduce the incidence of POCD in patients [8]. We do not know the exact reason behind this, but it seems that EA pre-conditioning/pretreatment (EA administered before/during surgery) provides more beneficial effects than post-treatment. 

An encouraging result from our study is that the neuroprotective effects of EA are only slightly lower than those of ibuprofen (Figure 8a,d,f). Although ibuprofen can provide a remarkable effect on POCD, which is consistent with our previous findings [10], ibuprofen may not be suitable for long-term use in elderly patients. Our data suggest that EA can be a potential alternative for those who had POCD but are not suitable for taking the anti-inflammatory drug.

Another significant aspect of our study was the investigation of the effects of EA on the later stages of POCD. We examined the cognitive and pathological changes in mice during both the early (7 days) and middle (14 days) phases of POCD. This approach provides a more comprehensive understanding of the pathological progression of this condition. The modulation of the immune response, either in the peripheral or central nervous system, has been frequently proposed as a key effect of EA [40,42,43,47]. In the present study, we focused on the long-term immunomodulatory effects of EA. We speculated that EA might protect the brain by suppressing peripheral (systemic) inflammation. Many research groups have reported that EA can reduce serum levels of IL-1β, IL-6, and TNF-α in patients after surgery [8]. However, we did not observe a drastic elevation of cytokine levels in the plasma during the early and middle postoperative periods on days 7 and 14. Only IL-6 and IL-10 levels were slightly elevated on day 7 (Appendix A). We cannot exclude the possibility that laparotomy induces only transient systemic inflammation, which can only be detected at an early time point. We then examined neuroinflammation. We found morphological changes in microglia and astrocytes in the brain on postoperative day 14, suggesting persistent neuroinflammation in these animals. EA attenuated these morphological changes in microglia and astrocytes, which may reflect a reduction in neuroinflammation (Figure 7). Interestingly, we found only a mild increase in the mRNA levels of TNF-α and IL-10 in the hippocampus on day 14, while the levels of other detected cytokines remained relatively unchanged. In the frontal cortex, there were no changes in the cytokine levels on days 7 and 14 (Figure 6). Together with the morphological changes in microglia and astrocytes, we propose that EA may protect the brain partly by suppressing neuroinflammation; however, it is likely that other mechanisms are also involved. For example, a recent study suggested that acupuncture together with ultrasound stimulation may affect local tissue oxidation reactions and membrane permeabilization and cause an increase in metabolism. Therefore, further research is needed to clarify the mechanisms of acupuncture [48].

Classical markers for AD include amyloid-β (Aβ) and tau proteins. In the clinical setting, changes in Aβ and tau levels after surgery appear transient. Previous research has discovered a notable increase in serum Aβ levels among older individuals on the first day of surgery, which then gradually diminishes and reverts to its baseline value by the third day following surgery [49,50]. Several studies reported that the serum level of tau increased significantly after surgery and remained elevated after 48 h [51,52]. Another study by Wiberg et al. showed similar results, and the group suggested that tau may possess a moderate ability to predict POCD after three months [53]. There is less clinical evidence to support long-term changes in tau levels in the peripheral circulation and brain after surgery. However, recent studies in animals have revealed that tau pathology, including acetylation, phosphorylation, and mislocalization, in the brains of animals persists for more than 2 weeks after surgery [10,54,55]. Our data showed that EA suppressed tau phosphorylation at 14 days after surgery (Figure 3). This finding is consistent with other reports that EA can improve cognition and reduce levels of phosphorylated tau in animal models of AD [56,57,58]. Moreover, EA also suppressed the phosphorylation of tau kinases GSK3β and JNK (Figure 4), which have been suggested as therapeutic targets for AD [59,60]. Taken together, our findings support the use of EA in the middle phase of POCD, and its effects may be partly through the modulation of tau.

This study had several limitations. First, laparotomy was used as a model for major surgery to investigate the effects of EA. Laparotomy involves physical stimulation of the intestine (i.e., general rubbing for 1 min), which mimics tissue disturbance during major surgery. However, since the procedures, level of traumatic injury, and pathological mechanisms vary among different types of operations, our findings should not be generalized and applied to other POCD models. For example, ischemia–reperfusion surgery may involve the generation of much higher levels of reactive oxidative species (ROS). The effects of EA in this condition may not be the same as those observed in our study. Second, with limited time and manpower, we did not explicitly investigate the involvement of ROS, anti-oxidative enzymes, and blood–brain barrier disturbance in our experiment. These have been reported as potential mediators modulated by EA, and the effects of EA have been reported by other researchers [61,62]. Third, while EA is a common technique in clinical practice, manual acupuncture without the use of electricity is frequently used by Chinese medicine practitioners. We chose EA but not manual acupuncture because the EA procedure was more repeatable with less interpersonal variation. Future studies may also use manual acupuncture to compare its effects with those of EA.

## 5. Clinical Implications

Based on our findings and corroborated by clinical and preclinical studies, we propose the following considerations for acupuncture in the management of POCD:(i)Acupuncture administered pre- or post-surgery demonstrates effectiveness in managing POCD.(ii)For patients unable to receive acupuncture before or during surgery, it remains a viable option following recovery from anesthesia and the return of consciousness.(iii)Postoperative acupuncture may require additional sessions to achieve comparable cognitive improvements to pre-operative treatments.(iv)Acupuncture may serve as a viable alternative to non-steroidal anti-inflammatory drugs for managing postoperative complications such as POCD.

## 6. Conclusions

The present study offers evidence that EA administered after surgery can attenuate cognitive impairment and some neuropathological changes in the brain. The protective effect of EA was slightly lower than that of ibuprofen, but it was still remarkable. EA attenuates the activation of microglia and astrocytes in the brain. However, mechanisms other than the suppression of inflammation, both peripherally and in the brain, may also exist and explain the observed benefits. Postoperative EA is a potential non-pharmacological intervention for the management of POCD.

## Figures and Tables

**Figure 1 biomolecules-14-01274-f001:**
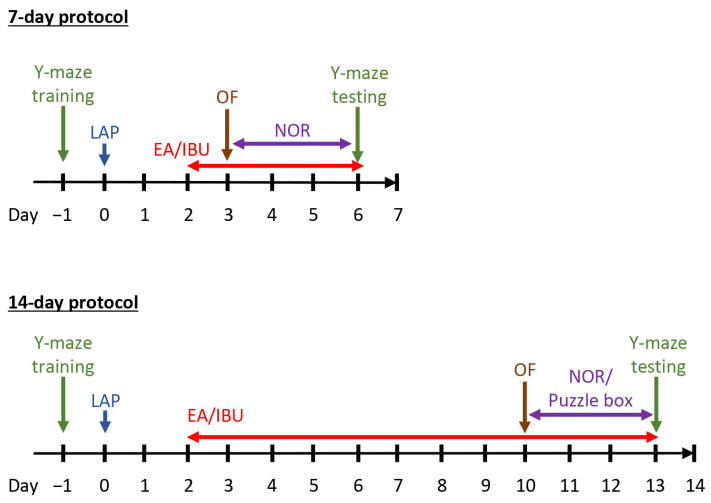
Timeline of experimental procedures.

**Figure 2 biomolecules-14-01274-f002:**
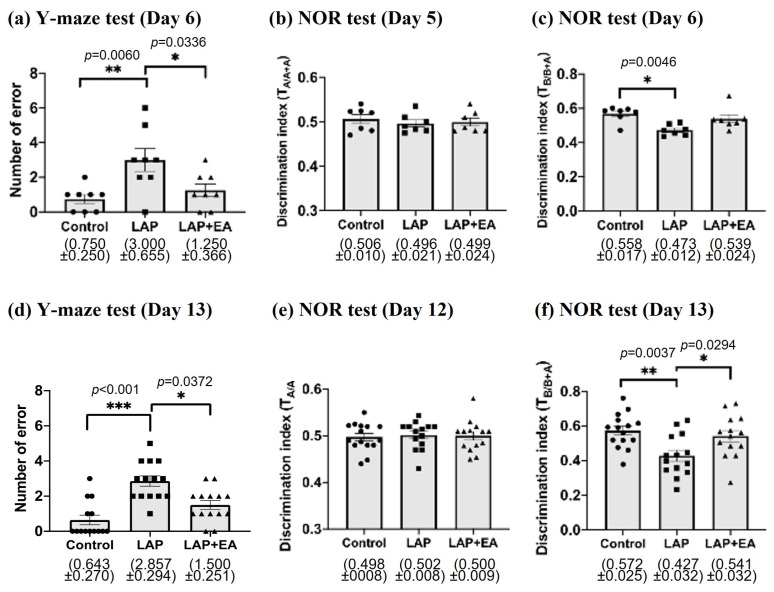
Electroacupuncture attenuated the cognitive impairment induced by laparotomy. The Y-maze test was used to assess associated memory at (**a**) 6 days (*n* = 8) and (**d**) 13 days (*n* = 14) after laparotomy. An increase in the number of errors made by the mice suggests cognitive impairment. The novel object recognition (NOR) test was used to assess hippocampal-dependent memory in the (**b**,**c**) early (*n* = 7) and (**e**,**f**) middle postoperative periods (*n* = 14–15). The data on days 5 and 12 in the NOR test showed that the mice had no preference among the two identical subjects placed in specific positions. The higher the discrimination index (TB/B+A), the better the novel object recognition ability of mice. Data were obtained from 2 to 3 batches of animals. Data were analyzed using one-way analysis of variance (ANOVA) followed by Tukey’s post hoc test for (**a**,**f**) and Kruskal–Wallis post hoc test for (**c**,**d**). * *p* < 0.05, ** *p* < 0.01, *** *p* < 0.001.

**Figure 3 biomolecules-14-01274-f003:**
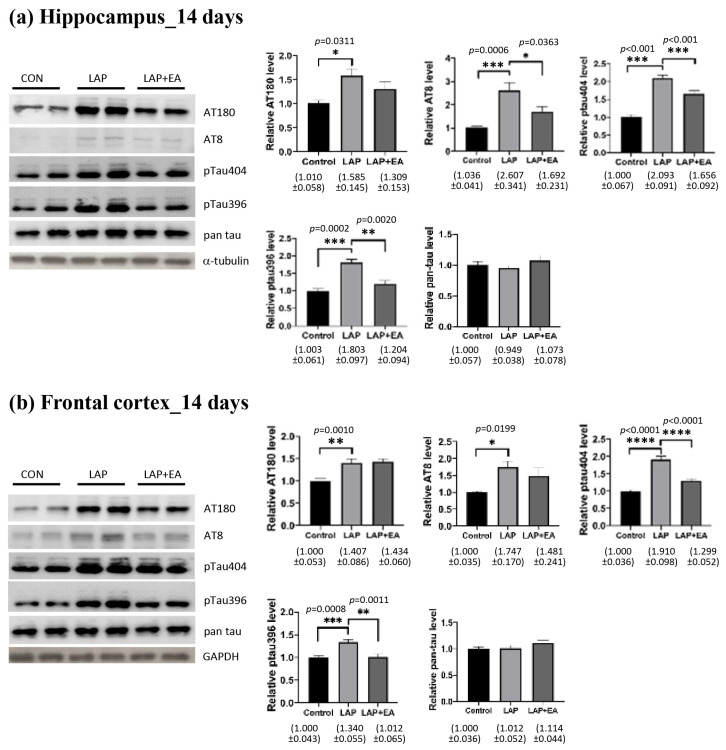
Electroacupuncture attenuated laparotomy-induced tau phosphorylation. Western blot analysis of changes in tau phosphorylation in the (**a**) hippocampus and (**b**) frontal cortex 14 days after laparotomy. Tau at different phosphorylation sites recognized by antibodies AT180 (pThr231/Ser235), AT8 (pSer202/Thr205), pTau404 (pSer404), and pTau396 (pSer396), and total tau data, were analyzed using one-way ANOVA followed by Tukey’s post hoc test for all the data. *n* = 7–8 for each group obtained from two batches of animals. * *p* < 0.05, ** *p* < 0.01, *** *p* < 0.001, **** *p* < 0.0001.

**Figure 4 biomolecules-14-01274-f004:**
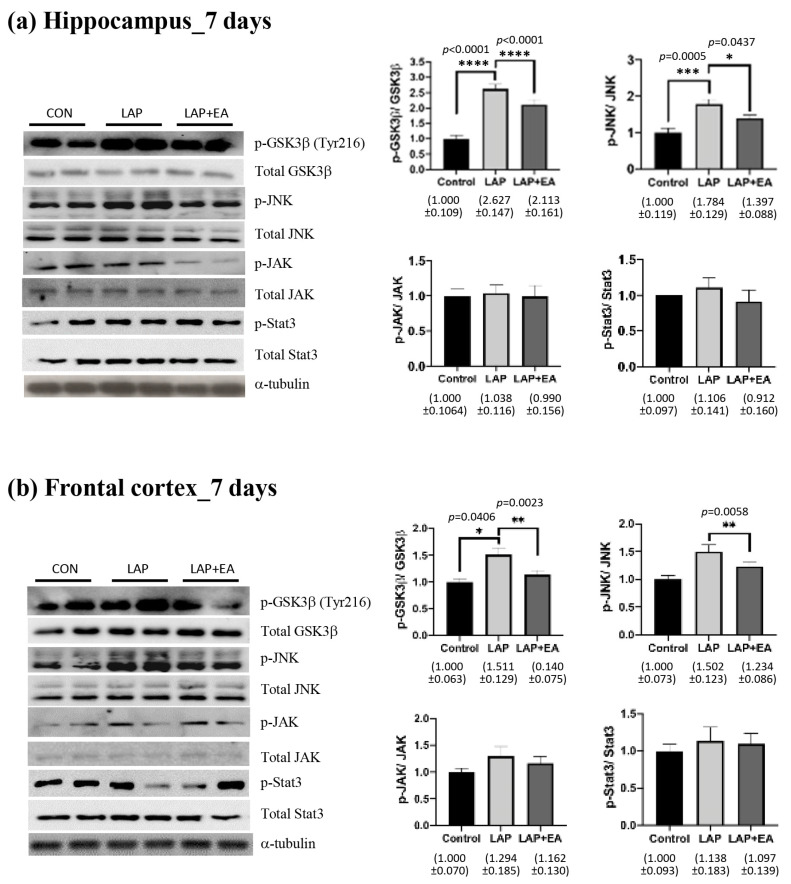
Electroacupuncture attenuated the activation of kinases. Western blot analysis of the expression of kinases in the (**a**) hippocampus and (**b**) frontal cortex in different groups 7 days after laparotomy. The image shows representative protein bands. Band intensities were measured using ImageJ, normalized to that of α-tubulin, and then analyzed using one-way ANOVA followed by Bonferroni’s post hoc test. *n* = 6–8 for each group. For all analyses, * *p* < 0.05, ** *p* < 0.01, *** *p* < 0.001, **** *p* < 0.0001.

**Figure 5 biomolecules-14-01274-f005:**
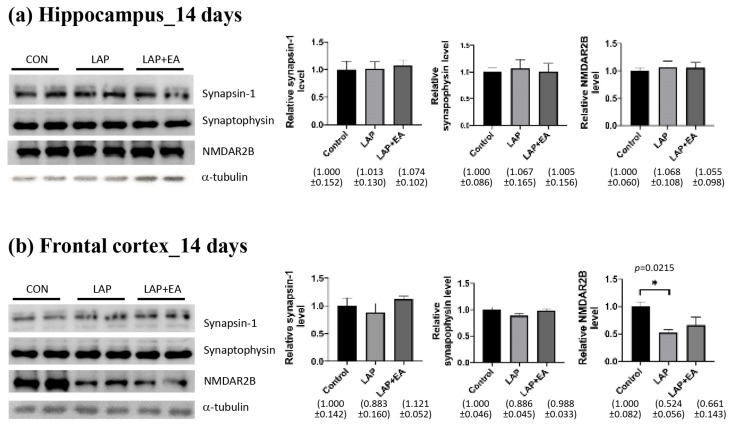
Electroacupuncture had no effect on laparotomy-induced changes in synaptic proteins. Western blot analysis of the expression of synaptic proteins (synaptin-1, synaptophysin, and NMDAR2B) in the synaptosome fraction of the mouse (**a**) hippocampus and (**b**) frontal cortex 14 days after laparotomy. Representative protein bands are shown. Band intensities were measured using ImageJ, normalized to that of α-tubulin, and analyzed using one-way ANOVA followed by Tukey’s post hoc test. *n* = 5–7 for each group. For all analyses, * *p* < 0.05.

**Figure 6 biomolecules-14-01274-f006:**
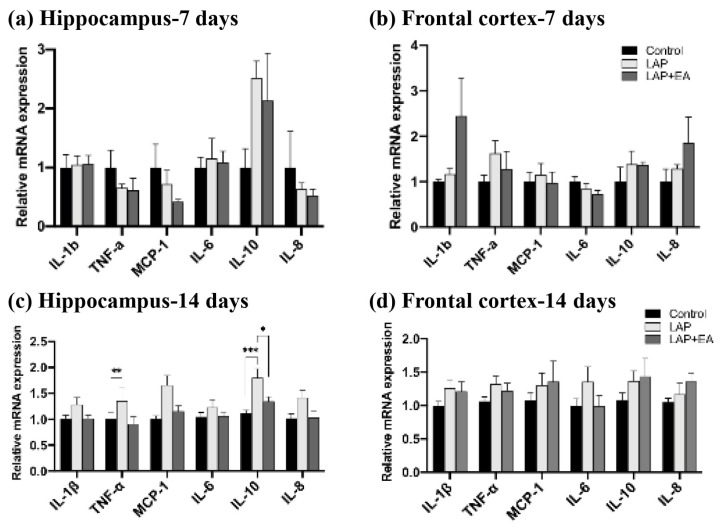
Levels of inflammatory cytokines after laparotomy. Neuroinflammation was investigated by detecting mRNA expression of pro-inflammatory cytokines in (**a**) the hippocampus and (**b**) the frontal cortex 7 days after laparotomy. *n* = 4–5. The mRNA expression levels of pro-inflammatory cytokines in (**c**) the hippocampus and (**d**) the frontal cortex 14 days after laparotomy. Data were analyzed using one-way ANOVA followed by Bonferroni’s post hoc test. *n* = 8–14. * *p* < 0.05, ** *p* < 0.01, *** *p* < 0.001.

**Figure 7 biomolecules-14-01274-f007:**
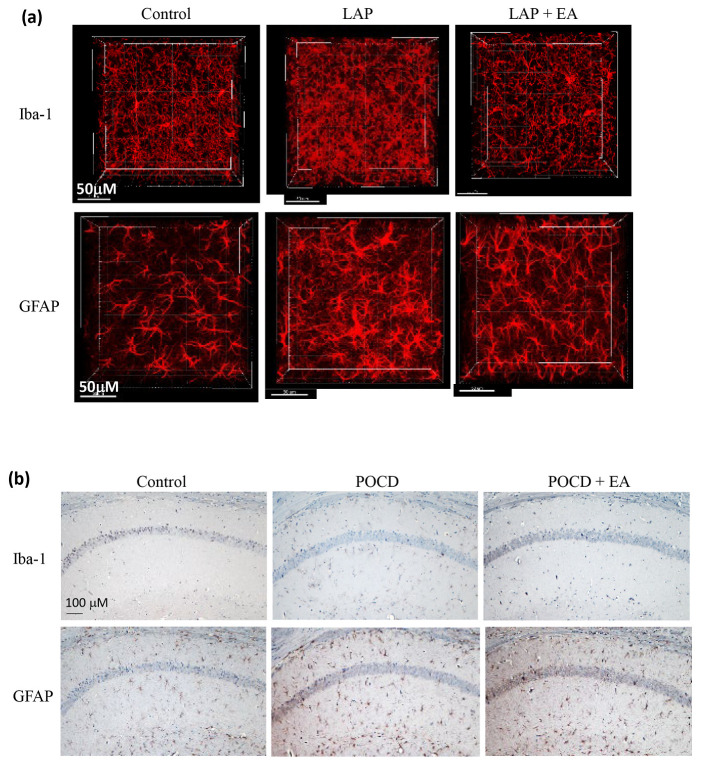
EA suppressed the activation of microglia and astrocytes after laparotomy. Immunohistochemical staining of brain slides for microglia (Iba-1) and astrocytes (GFAP) was performed using the (**a**) Accu-OPTICLear tissue-clearing technique. After all the behavioral tests, mice were sacrificed on day 14, and their brains were embedded for immunohistochemical staining using the transparent brain protocol, which allows better study of the 3D morphology of the target cells. Magnification = 40×. Scale bar = 50 μM. Images were taken from the CA1 region of the hippocampus for Iba-1-positive microglia and GFAP-positive astrocytes. Data represent 4 mice in each group. The results were confirmed using the (**b**) conventional ABC immunohistochemical staining method. Images were obtained from the hippocampal CA1 region. Data represent 3–4 mice (*n* = 3–4). Magnification = 10× Scale bar = 100 μM.

**Figure 8 biomolecules-14-01274-f008:**
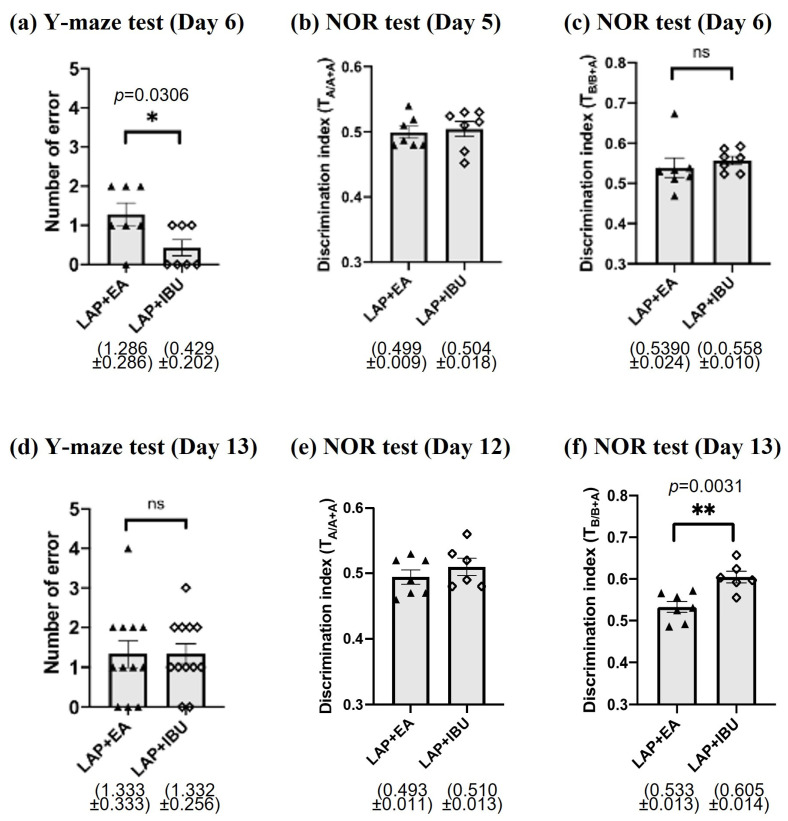
A comparison of cognitive performance between the EA and ibuprofen groups using Y-maze and NOR tests. The Y-maze test was used to assess associated memory (**a**) 6 days (*n* = 7) and (**d**) 13 days (*n* = 12) after laparotomy. The novel object recognition (NOR) test was used to assess hippocampal-dependent recognition memory in the (**b**,**c**) early (*n* = 7) and (**e**,**f**) middle period (n = 6–7) after laparotomy. Data were obtained from 2 to 3 batches of animals. Data were analyzed using an unpaired Student’s *t*-test. * *p* < 0.05, ** *p* < 0.01, ns = not significant.

**Figure 9 biomolecules-14-01274-f009:**
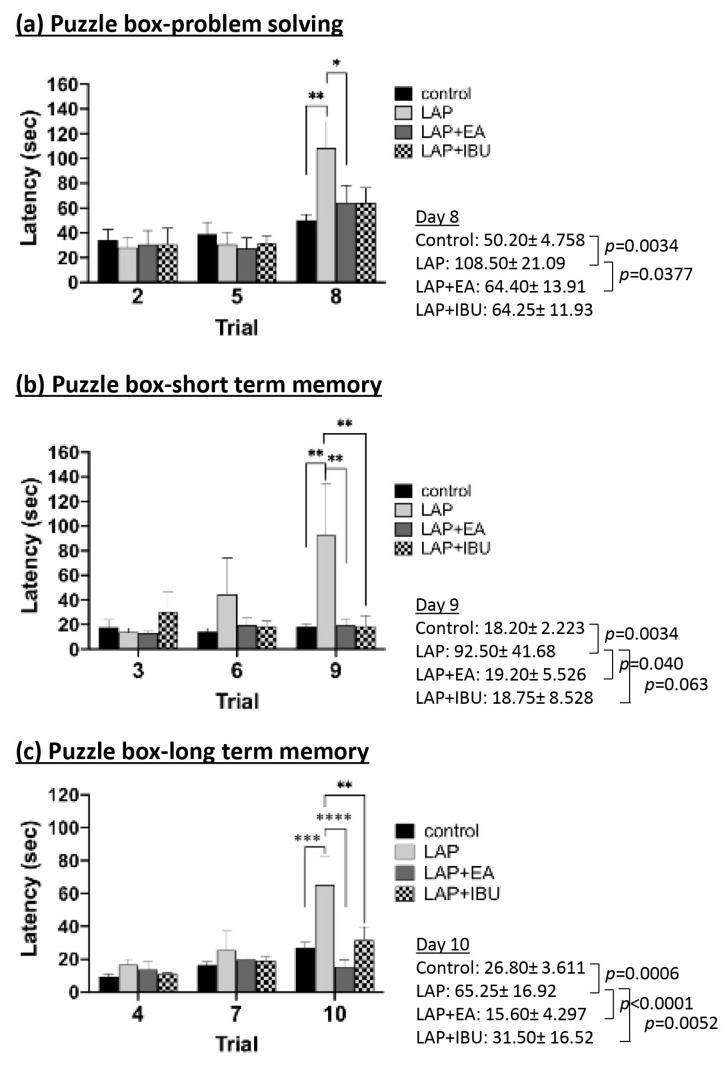
The investigation of executive function and cognitive ability in the middle postoperative period using the puzzle box test. An individual mouse was placed in a two-compartment arena. The experiments were conducted on days 10–13. The time taken to enter the goal zone was recorded. During this period, the entrance to the goal zone was blocked with different objects with increasing difficulty. Hence, the increased trial number in each figure represent the increased level of challenges to the mice. (**a**) Problem-solving ability was investigated in Trials 2, 5, and 8. (**b**) Short-term memory was investigated in Trials 3, 6, and 9. (**c**) Long-term memory was investigated in Trials 4, 7, and 10. Data were analyzed using two-way ANOVA followed by Turkey’s post hoc test. *n* = 6, * *p* < 0.05, ** *p* < 0.01, *** *p* < 0.001, **** *p* < 0.0001.

## Data Availability

All data presented in the manuscript and/or Appendix A are available upon request.

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
