# Peer review of "Postoperative Electroacupuncture Boosts Cognitive Function Recovery after Laparotomy in Mice"

_biomolecules, 2024, doi:10.3390/biom14101274_

Round 1
Reviewer 1 Report
Comments and Suggestions for Authors
The manuscript titled “Postoperative Electroacupuncture Boosts Cognitive Function Recovery after Laparotomy in mice” presents a study investigating that electroacupuncture (EA) could reduce inflammation and cognitive dysfunction induced by laparotomy over a longer period. The authors have conducted a series of experiments to evaluate the postoperative EA on cognitive changes. They found that Electroacupuncture attenuated laparotomy-attenuated tau phosphorylation & activation of kinases. Overall, the study is well-designed, interesting and the results are presented clearly.
1) The introduction needs to be strengthened. Please add more studies related to EA.
2) Can authors explain the entire study in the form of a graphical abstract.
3) Why did the authors choose only male mice for their study?
4) Please add magnification and scale bars in Figure 8b.
Author Response
- The introduction needs to be strengthened. Please add more studies related to EA.
Response: Thank you for your comment, we agree with this. Therefore, we have revised our revision and add more information in the introduction section. Please refer to our updated manuscript page 4 (changes have been highlighted).
- Can authors explain the entire study in the form of a graphical abstract.
Response: Thank you for your suggestions. We have added a graphical abstract on page 1 bottom (after the abstract).
- Why did the authors choose only male mice for their study?
Response: Thank you for pointing this out. We chose to use only male mice because previous studies have reported gender differences in spatial learning, memory, and hippocampal plasticity (1-3). In this pre-clinical study, we aimed to minimize the influence of gender factors and focus on the therapeutic effects of acupuncture. Therefore, we selected male mice as our study subjects. We have added this justification in Section 2.1, Animal Husbandry (page 3, line 122)
- Please add magnification and scale bars in Figure 8b.
Response: Thank you for pointing this out. We have added the magnification and scale bar on Figure 8b (page 14).
Reviewer 2 Report
Comments and Suggestions for Authors
Thanks to the authors for this very interesting research on the assessment of using electroacupuncture for the reduction of postoperative cognitive dysfunction after laparotomy in mice.
After a thoughtful reading of the paper, I would like the authors addressing the following:
1. Line 20: authors mention that “previous studies have shown that acupuncture reduces POCD within 7 days of surgery”. Then they go into mention that “The long-term effects of acupuncture are unknown”. They also follow on hypothesising that electroacupuncture could reduce POCD for a longer period. How can authors argue about this reduction if the long-term effect of the conventional treatment is unknown?
2. Line 97 (Section 2: Materials and Methods): Authors used twelve mice and created three experimental groups of four mice each. From the statistical point of view (despite of what is mentioned in line 109), could the authors argue as to why such a small sample size could lead to reliable results?
3. Line 281: “Our findings indicate that all three groups exhibited similar levels of these parameters (…)”. Authors cannot argue similarities, only differences (statistically speaking).
4. I am puzzled as to why authors use different number of mice for the 7-day test (8 per group) vs 14-day test (14 per group). In addition, how these number match with observation in point 2 above?
5. Following the tests mentioned in 4 above, why the authors did not run a repeated-measures ANOVA? (as this seems a longitudinal study?)
6. Similar observations as 5 above for tests presented in Sections 3.3 to 3.5. All of them seem longitudinal studies.
7. It is hard to understand how the authors conducted the tests in section 3.6, as the number of experimental subjects (mice) changes. For instance, in the Y-Maze test the number of mice is 7 in day 6; whereas the number of mice changes to 12 in day 14. How would the authors expect feasible and reliable statistical conclusions under these conditions?
8. Paragraph describing Figure 10 needs grammatical revision.
9. Can the authors explain in which procedures / tests the following statistical techniques were implemented? (Section 263): Mann-Whitney test, Dunn’s post-hoc test, Kruskal Wallis test, and Dunn’s post-hoc test? Where are the values of the means and standard errors?
10. Please note that my main concern in this paper is the lack of statistical rigour. I could not find a single metric (besides broad p-values) to back up authors’ findings. That is, F-Scores, degrees of freedom, confidence intervals, power measurements, and analysis of effect sizes are all missing. If the authors which to publish this research, this point needs to be addressed thoughtfully.
Comments on the Quality of English Language
Please, some revisions are needed.
Author Response
- Line 20: authors mention that “previous studies have shown that acupuncture reduces POCD within 7 days of surgery”. Then they go into mention that “The long-term effects of acupuncture are unknown”. They also follow on hypothesising that electroacupuncture could reduce POCD for a longer period. How can authors argue about this reduction if the long-term effect of the conventional treatment is unknown?
Response: Thank you for pointing this out. We have revised our writing in the abstract for clarity: “…Previous studies have shown that acupuncture improves cognition in the early phase of POCD…” (Page 1, line 23).
Currently, there are no approved drugs for treating POCD once it develops, so research has focused on prevention rather than treatment. We have added lines to the introduction to emphasize the significance of developing POCD treatments: “Moreover, there are currently no approved medications for the treatment of POCD, although some clinical reports suggest that the use of lidocaine (a local anesthetic) or parecoxib (a nonsteroidal anti-inflammatory drug) may reduce the incidence of POCD. Therefore, it is essential to investigate and develop effective management strategies to help patients restore their cognitive function at different phases after surgery.” (Page xxx, line xxx)..” (Page xxx, line xxx).
- Line 97 (Section 2: Materials and Methods): Authors used twelve mice and created three experimental groups of four mice each. From the statistical point of view (despite of what is mentioned in line 109), could the authors argue as to why such a small sample size could lead to reliable results?
Response: It seems there might be some misunderstanding. In our manuscript (line 120), we stated that "Twelve-week-old male C57BL/6N mice, weighing 25 ± 3 g, representing early adulthood, were purchased from…". This provides the age, weight, and general conditions of the mice. In Section 2.2, Grouping of Animals (page 3, line 144), we stated that “the mice were randomly divided into three groups: (1) sevoflurane alone (control), (2) laparotomy alone, and (3) laparotomy plus EA”. This indicates the grouping for the initial phase of our experiment. We did not specify the total number of mice in each group in this section, as the numbers vary for each test. As shown in Figures 3, 4, 5, and 6, at least 7 mice were used in each group, which ensures statistical significance and reliable results.
- Line 281: “Our findings indicate that all three groups exhibited similar levels of these parameters (…)”. Authors cannot argue similarities, only differences (statistically speaking).
Response: Thank you for pointing this out. We have revised the sentence for clarity: "There was no significant difference between the three groups in terms of the total distance travelled and time spent in the outer zone, indicating no differences in anxiety- or depression-like behavior (page 8, line 314).”
- I am puzzled as to why authors use different number of mice for the 7-day test (8 per group) vs 14-day test (14 per group). In addition, how these number match with observation in point 2 above?
Response: I apologize for the lack of clarity in our original manuscript. As clarified in point 2, we used a minimum of 7 mice for each test. Different numbers of mice were used for the 7-day and 14-day tests to minimize animal use in accordance with ethical and welfare principles. We utilized different batches of animals for the 7-day and 14-day protocols. This was necessary because we collected both behavioural and biochemical outcomes at these time points, requiring mice to be sacrificed for sample collection. To clarify this, we have revised our manuscript to include: "Based on our electroacupuncture protocol described in Section 2.4, mice were entered into either the 7-day or 14-day protocol and were sacrificed on day 7 or day 14, respectively." (page 4, line 146).
5. Following the tests mentioned in 4 above, why the authors did not run a repeated-measures ANOVA? (as this seems a longitudinal study?)
Response: I apologize for the lack of clarity. As mentioned in our response to comment 4, different batches of mice were used for the 7-day and 14-day protocols. Since the measurements were not taken from the same animals at different time points (i.e., this is not a longitudinal study), we cannot run a repeated-measures ANOVA.
6. Similar observations as 5 above for tests presented in Sections 3.3 to 3.5. All of them seem longitudinal studies.
Response: I apologize for the lack of clarity. As mentioned in our response to comment 4, different batches of mice were used for the 7-day and 14-day protocols. Since the measurements were not taken from the same animals at different time points (i.e., this is not a longitudinal study), we cannot run a repeated-measures ANOVA.
7. It is hard to understand how the authors conducted the tests in section 3.6, as the number of experimental subjects (mice) changes. For instance, in the Y-Maze test the number of mice is 7 in day 6; whereas the number of mice changes to 12 in day 14. How would the authors expect feasible and reliable statistical conclusions under these conditions?
Response: Thank you for your question. As mentioned in our response to comment 4, different numbers of mice were used for the 7-day and 14-day tests to minimize animal use in accordance with ethical guidelines. We used different batches of animals for these protocols. Due to limitations in cage availability and space, we conducted the experiments in several batches. For example, in batch one, we conducted the 7-day protocol with 3-4 mice per group. After analyzing the data, if statistical significance was achieved (as in Figure 8a for the Y-maze test at 7 days), no further experiments were conducted. If more variation was observed, additional batches were run to confirm findings, resulting in more mice being used for the Y-maze test at 14 days (Figure 8d).
8. Paragraph describing Figure 10 needs grammatical revision.
Response: Thank you for your suggestion. We have revised the grammar of the figure legend for Figure 10. Changes have been highlighted (now figure 9, page 16).
9. Can the authors explain in which procedures / tests the following statistical techniques were implemented? (Section 263): Mann-Whitney test, Dunn’s post-hoc test, Kruskal Wallis test, and Dunn’s post-hoc test? Where are the values of the means and standard errors?
Response: Thank you for your suggestions, we have added the information about the post-hoc test that we used in the figure legend . This can avoid over-information which make the reading of manuscript difficult. For the means and standard error, we have added these information in the figures.
- Please note that my main concern in this paper is the lack of statistical rigour. I could not find a single metric (besides broad p-values) to back up authors’ findings. That is, F-Scores, degrees of freedom, confidence intervals, power measurements, and analysis of effect sizes are all missing. If the authors which to publish this research, this point needs to be addressed thoughtfully.
Response: Thank you for your suggestion. We acknowledge the importance of statistical analysis and have ensured our findings are accurately reported. In preclinical studies, it's uncommon to include F-scores, degrees of freedom, confidence intervals, power measurements, and effect sizes, as these are more typical in clinical research, such as trials or surveys. This is due to:
- Smaller Sample Sizes: Animal studies often have smaller sample sizes due to ethical and logistical constraints, limiting the applicability of certain statistical measures like power and effect size.
- Standard Practices: The focus is traditionally on p-values to assess significance in these studies.
Additionally, unlike clinical trials, preclinical studies investigate multiple behavioral and biological outcomes without a predefined primary outcome, making it challenging to calculate effect sizes. We provide means, standard errors, and p-values, which are standard in preclinical research. We appreciate your feedback and will consider ways to enhance the statistical rigor where feasible.
Reviewer 3 Report
Comments and Suggestions for Authors
The authors have developed a well-conducted and well-written study aimed to evaluate the potential of electroacupuncture as a viable non-pharmacological intervention for managing postoperative cognitive dysfunction (POCD).
However, I would like to make a few observations. Please address all comments in the above sections and as follows:
1. Based on the provided information, it cannot be confirmed whether the sample size was calculated using statistical methods. Please, I ask the authors to provide the sample size calculation performed.
2. Statistical analyses were conducted using various tests, including ANOVA and t-tests, to determine the significance of the results, but it does not specify any calculations related to statistical power.
3. There is also a biochemical explanation to electroacupuncture that I recommend the authors to add in their development, using the following article: DOI: 10.52586/5017
4. In the Methodology section, all the data of the start and end dates of the study are missing.
5. Could you add a section on "Clinical Implications"?
6. There are too many figures in the manuscript. I recommend that the essential figures remain in the manuscript, and the rest as supplementary material. On the contrary, I recommend that some tables be included.
7. Add the type of study in the title and in the methodological section.
8. I advise the authors to give a clinical touch to their work by commenting on the following clinical research paper in the light of their results: DOI: https://doi.org/10.3390/ijerph20032617
Comments on the Quality of English LanguageNo comments
Author Response
- Based on the provided information, it cannot be confirmed whether the sample size was calculated using statistical methods. Please, I ask the authors to provide the sample size calculation performed.
Response: Thank you for your suggestion. We have added information in Section 2.9 (page 7, line 293) about the sample size calculation:
“The number of rats in each treatment group was calculated using the equation: N = 1 + 2C(s/d)2 from the Institute for Laboratory Animal Research (U.S.) Committee on Guidelines for the Use of Animals in Neuroscience and Behavioral Research [Power (1-β) of the treatments = 80%; significant level a = 5%, C = 7.85].”
Please note that sample size calculation is an estimation at the proposal stage and may vary during the experiment. We typically stop further experiments when statistical significance is achieved, reducing the number of animals used and complying with the National Institutes of Health's Guide for the Care and Use of Laboratory Animals.
- Statistical analyses were conducted using various tests, including ANOVA and t-tests, to determine the significance of the results, but it does not specify any calculations related to statistical power.
Response: Thank you for your suggestion. Please refer to our response in your comment 1. We have added these information in Section 2.9 (page 7, line 293).
- There is also a biochemical explanation to electroacupuncture that I recommend the authors to add in their development, using the following article: DOI: 10.52586/5017
Response: Thank you for your suggestion and the updated literature. We have incorporated this into our discussion to support the explanation of acupuncture’s mechanism.
“For example, a recent study suggested that acupuncture together with ultrasound stimulation may affect local tissue oxidation reactions and membrane permeabilization and cause an increase in metabolism. Therefore, further research is needed to clarify the mechanisms of acupuncture (1).” (page 18, line 556)
- In the Methodology section, all the data of the start and end dates of the study are missing.
Response: Thank you for your comment. Our experiments were conducted in multiple batches with varying start and end dates for each behavioral outcome, and biochemical tests were performed throughout the study period. Therefore, it's not feasible to provide specific start and end dates for individual experiments. Additionally, it is uncommon to include this information in animal studies. We appreciate your understanding.
- Could you add a section on "Clinical Implications"?
Response: Thank you for your suggestion. We agree that adding information about clinical implications would provide more insight to our audience. We have added this section after the discussion:
“Clinical implications
Based on our findings and corroborated by clinical and preclinical studies, we propose the following considerations for acupuncture in the management of POCD:
- Acupuncture administered pre- or post-surgery demonstrates effectiveness in managing POCD.
- For patients unable to receive acupuncture before or during surgery, it remains a viable option following recovery from anesthesia and the return of consciousness.
- Postoperative acupuncture may require additional sessions to achieve comparable cognitive improvements to pre-operative treatments.
- Acupuncture may serve as a viable alternative to non-steroidal anti-inflammatory drugs for managing postoperative complications such as POCD.” (page 19, line 594)
- There are too many figures in the manuscript. I recommend that the essential figures remain in the manuscript, and the rest as supplementary material. On the contrary, I recommend that some tables be included.
Response: We agree that there are quite a lot of figures in the manuscript. We have moved the one about Body weight, general locomotor activities levels and anxiety- or depression-like behaviors (original Figure 2) to the supplementary figures.
- Add the type of study in the title and in the methodological section.
Response: Thank you for your suggestion. We have indicated in the title and methodological section that mice were used as subjects in our study. Unlike clinical studies, we are not conducting an RCT or using a quasi-experimental design. It is also uncommon to indicate blinding in the title for animal studies. We would appreciate clarification on the specific information the reviewer would like us to include in the title and method section.
- I advise the authors to give a clinical touch to their work by commenting on the following clinical research paper in the light of their results: DOI: https://doi.org/10.3390/ijerph20032617
Response: Thank you for your suggestion and the updated literature. We have incorporated this into our introduction to support the potential application of acupuncture on patients with surgery. (page 3, line 108)
Round 2
Reviewer 1 Report
Comments and Suggestions for Authors
The authors have incorporated all the suggestions in the revised manuscript.
Reviewer 2 Report
Comments and Suggestions for Authors
Thanks once again to the authors for considering my observations, comments, and questionings. I appreciate the rigour and the time spent in updating the paper accordingly.
No extra comments from my side.
Reviewer 3 Report
Comments and Suggestions for Authors
The authors have improved with the current version the previous version of their manuscript, so I recommend its publication.
Congratulations
Comments on the Quality of English LanguageNo comments.